# Maturity Model for Sustainability Assessment of Chemical Analyses Laboratories in Public Higher Education Institutions

Marco Antonio Souza [1,*], Mario Orestes Aguirre González [1] and André Luís Santos de Pinho [2]

1   Department of Production Engineering and Graduate Program in Petroleum Science and Engineering, Federal University of Rio Grande do Norte Federal, Natal 59078-970, Brazil; mario.gonzalez@ufrn.br

2   Department of Statistics, Federal University of Rio Grande do Norte, Natal 59078-970, Brazil; andre.pinho@ufrn.br

*   Correspondence: marco.souza@ufrn.br

**Abstract:** The increasing demand for sustainable products and services has become a constant requirement for the different stakeholders in an organization. Higher Education Institutions (HEI) have a crucial role in training conscious and competent professionals to lead these changes. Chemical analyses laboratories bring together the proper mix, where the adoption of mature and efficient management systems proves to be crucial not only to better train the institutions' human resources but also to cooperate in recruiting research projects and other services to benefit society. Maturity models assist in the needed stages for sustainable growth to take place. This paper proposes a maturity model based on standardized norms to guide adjustments related to quality, risks, safety, and environment of chemical analyses laboratories in public higher education institutions. This research was done in four stages: theoretical research about maturity models, sustainability, and integrated management systems; survey with laboratories; case study at a chemical analyses laboratory of an HEI; and structuring and validating a maturity model. As the main results, it was observed that more than 80% of public laboratory managers believe it would be relevant to adopt a maturity model to help organize the laboratory's internal and external processes. 86% of public laboratory managers understand that using management systems can contribute to hiring new services. We can also observe that 42.9% of public laboratory managers do not know any maturity model. As conclusion, the model includes eight dimensions, 31 subdimensions, and 204 management practices to assess and guide chemical analyses laboratories to sustainable maturity levels.

**Keywords:** integrated management systems; maturity model; sustainable development; laboratory; ISO 17025

## 1. Introduction

Ever since the first half of the 20th century, we have seen constant shifts on a global scale in the organization of work processes and organizational management. Economic, technological, philosophical, and social changes shift dominating paradigms, which demand the evolution of management systems. Management models evolve, and new concepts such as systemic vision, risk mindset, and sustainability gain prominence.

By integrated management system, we mean the joint use of two or more management systems based on normative standards to manage an organization's processes, reduce operational risks, potentialize opportunities, raise the organization's resilience and improve its performance, and contribute to the business's sustainability.

Amongst the reasons that lead organizations to use management systems in an integrated way are increasing performance [1], improving performance and internal communications [2,3] as well as compatibility between normative standards [4,5], increasing synergy between systems and eliminating redundancies [6–8], market gain [4,8], and meeting legal requirements [2,9,10].

Risk mentality is one of the main changes to the ISO standards review as of 2015. This concept is vital because it establishes a base to increase systems' efficacy [11]. ISO 31000 [12] defines a set of principles and guides to implement risk management. It states that its principles are the foundation for managing risks and that they are suitable for structuring risk management processes. Domingues [13] adds that an IMS should be accompanied by a risk management approach, given that this is the factor that integrates systems.

Continuous improvement should be part of the foundation of management systems. Tracking set objectives and goals allows strategic and operational diagnoses to be more assertive and allows perception of how the system is evolving. At each new cycle, the system should be more robust, undergoing failure review, correcting non-conformities, and making processes interdependent. At each new cycle, the culture matures and makes processes more long-lasting and sustainable.

Nunhes; Bernardo; Oliveira [14] identified possible contributions and gaps in the development of studies about integrated management systems; among the items identified were the need to investigate the impact of certified management systems on sustainable development, as well as the need to develop system integration proposals for the sustainability of corporations to optimize results related to sustainable development.

Integrating management systems such as quality, environment, health, and occupational safety contributes to sustainability [15]. However, to assess the evolution levels of a management system, it is necessary to adopt a maturity model. Maturity models allow for the analysis of incomplete evolution stages (in general, by organizations or processes) using multidimensional criteria [16]. It may be defined as a set of sequential levels that, together, would describe an anticipated pathway forward, that is desired and logical, from the initial to a final stage of maturity [17]. In this context, the literature has applied maturity models to various areas, such as business management, project management, knowledge management, culture assessment, software development, quality improvement, chemical industry, food industry, and safety [16,18–26].

Another important point regards higher education institutions. Research at the educational institution level has reasserted the role universities have in the training of professionals who are aware of their responsibility in their professional work to face challenges related to sustainability, safety, and quality of life of the researchers involved in the teaching–learning process and in the quality of life of society that is impacted by its processes [27,28].

Higher education institutions (HEIs) have a strategic role in sustainable development in the dimensions of teaching, research, dissemination, and management. They are also responsible for training professionals who are aware of their role in sustainability and for providing them with the aptitude and competencies necessary to address future challenges in this area. European universities are moving forward with the implementation of Agenda 2030, and a series of initiatives can be adopted by the HEIs to implement sustainability actions, such as implementing Sustainable Development Goals (SDGs) systematically; starting actions aligned to institutional documents and the strategic mission; integrating the SGDs in the curriculum and in learning; and using training to communicate with the university community about the topic [29].

In Brazil, public HEIs have a relevant role in research development and are responsible for 95% of the country's scientific production. However, because they are maintained and funded by the government, they are subject to budgetary restrictions that may impact their infrastructure or make ongoing research impracticable.

To strengthen research and innovation, the Brazilian government passed the Innovation Act (Law number 13.243/16) [30], which allows for cooperation between scientific and technological institutions and private companies. Among possible interactions, the phrasing of the law provisions laboratory, instrument, and material sharing as well as the use of facilities for research activities (Article 4). This includes service development (Article 9) according to the partnership [31].

Teaching and research laboratories are fundamental to a country's economic and social development. Laboratories belonging to universities and research institutes allow

theory and practice to be reconciled, allowing undergraduate and graduate students to experience, experiment with, and develop the necessary skills to build student and professional competencies.

In research carried out in Chinese companies, Zhang and Jin [32] observed that ESG management plays an important role in promoting innovation capacity in green technology. Among the implications considered in the study, investment in R&D for green technological innovation is mentioned, to promote and achieve its own high-quality development in the future. Jun Deng et al. [33] pointed out the need to invest in emergency management systems, which include emergency prevention, emergency preparedness, and emergency response.

In this regard, we highlight the importance of adequately equipping and managing laboratories in public HEIs to meet potential emerging demands services they may have. By adopting certified management systems, research laboratories would use the same document, organizational, and language standards that companies and private laboratories use, facilitating managerial processes between both entities.

In this context, the adoption of integrated management practices in laboratories aligned with the HEIs' strategic mission and seeking to expand sustainability gradually contributes to the integration of SDGs in the teaching-learning process, as well as corroborates to occasional bidding processes for partnership agreements between HEIs and private companies, and it promotes the awareness of technicians, professors, students, and of other members in the university community.

The bibliographical research to develop this type of work found only two papers associating maturity models and management systems with chemical analysis laboratories [34,35]. The models, however, were limited to providing a situational diagnosis of the maturity level of the implemented system, and the proposal of effective integration actions that may improve the maturity level was not part of their scope. The articles did not focus on public HEIs and failed to demonstrate the existence of a maturity model that would assist chemical analysis laboratories in the gradual and sustainable implementation of integrated management systems (IMS).

In view of the above, the following question was formulated: how can a maturity model assist chemical analyses laboratories in reaching higher sustainability levels and, with this, add to higher education institutions' (HEI) sustainability policies and actions?

In order to answer the research questions, the following propositions were formulated:

P1: Structuring a maturity model based on standardized management systems is relevant to managers and positively affects reaching sustainability at the laboratory level.

P2: HEIs' chemical analyses laboratory managers are interested in adopting standardized management systems to organize and improve their processes.

P3: Chemical analyses laboratories at universities and HEIs do not have structured management systems; they have isolated tools and/or methodologies to coordinate and control their operations and routines.

P4: It is possible to structure a maturity model suited to a chemical analyses laboratory context to achieve sustainable activities.

As the main results, it was observed that more than 80% of public and private laboratory managers believe that a maturity model would help to organize internal processes. In total, 86% of public laboratory managers understand that the use of management systems optimized laboratory processes and can contribute to hiring new services, which reveals how relevant this is. It was also possible to elaborate a maturity model to assess and gradually build an integrated management system for the sustainable development of chemical analyses laboratories.

This paper is divided into four sections. Section 1 contextualizes and clarifies the intended goals. Section 2 deals with the methodology used to accomplish this survey. Section 3 presents the results achieved and the discussions on the data collected to build the model. Section 4 presents conclusions observed based on the research propositions.

## 2. Materials and Methods

This is a descriptive exploratory survey in a combined approach (qualitative and quantitative). As to the methods, they were bibliography-based research [36], field research (survey), and case studies [37–39]. The methodology was divided into four steps: theoretical research, field research, case study, and maturity model structuring (Figure 1).

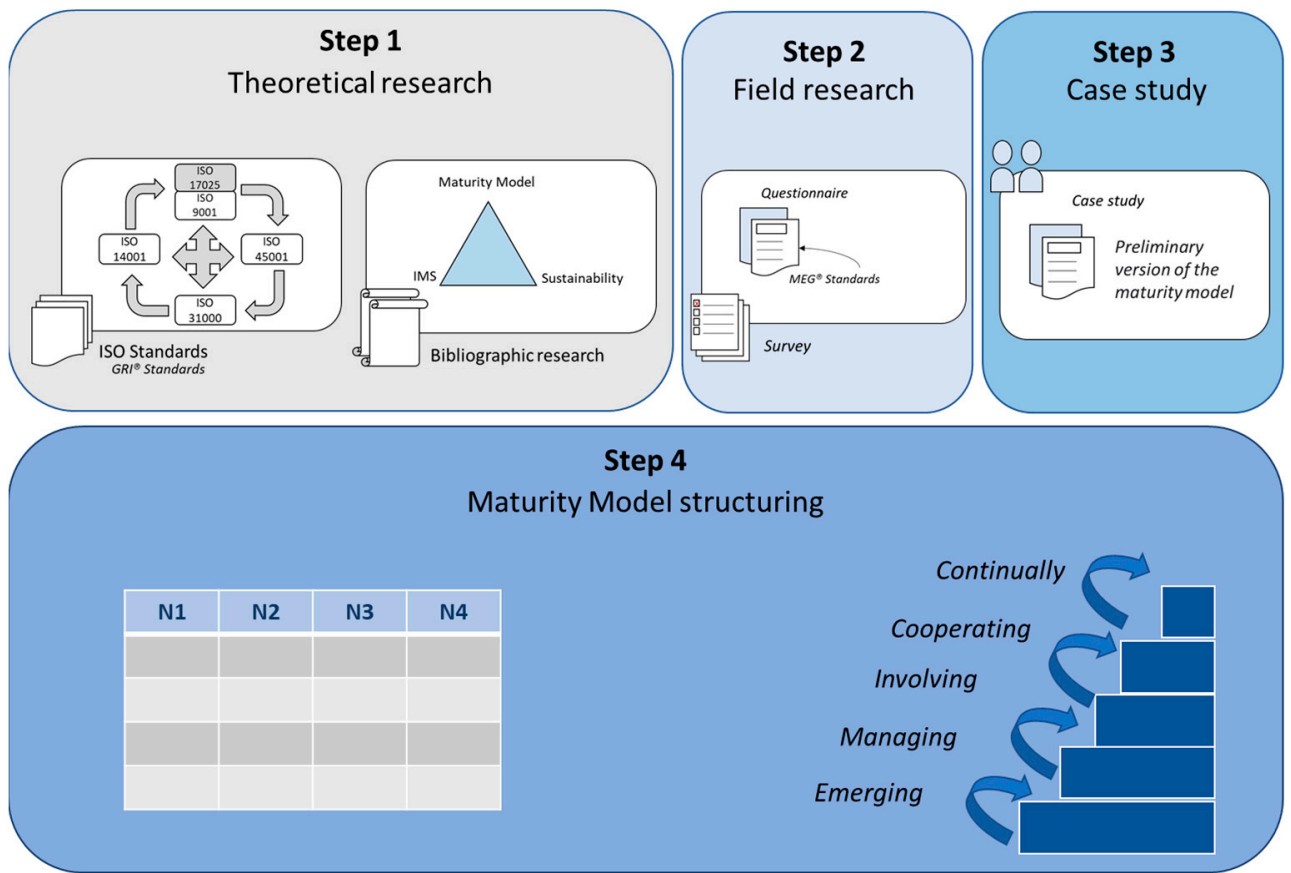

**Figure 1.** Research methodology [11,12,40–42].

In theoretical research, five stages of searches were conducted to unify three different yet interrelated topics in one single study: maturity model, sustainability, and integrated management systems applied to the laboratory management setting.

In the first step, we researched papers on maturity models associated with integrated management systems. Certifiable Quality standards (ISO 9001 [11]), Occupational Health and Safety (ISO 45001 [42] or OHSAS 18001 [43]), and Environment (ISO 14001 [41]) were used as search parameters, as well as maturity models developed and applied to the reality of these systems. The bibliographical research also considered concepts of Sustainability and Risk Management (ISO 31000 standard [12]) associated with constructing these models. The term "ISO 17025" was used as a search parameter for laboratory management, referring to the standard used in the certification of testing laboratories to ensure the quality of operations. Three scientific databases were used in the research: Emerald Insight, Science Direct, and Scopus. The first survey was carried out between December and January 2019, repeated between December and January 2021, and again in April 2023. Articles were searched between 1995 and 2023 for the words on the strings anywhere in the article (Table 1).

In the second stage, field research (survey) was carried out based on collecting institutional data from the laboratories, their management practices, and the perception of relevance for adopting management systems. Data were collected using a semi-structured

questionnaire with open-ended and closed questions, presented in Appendix A. The questionnaire was prepared using the Google Forms® platform, in which an explanatory email with an access link was sent to the respective managers. At the end of the questionnaire response period, the collected data were analyzed using statistical software R (version 4.2.2).

**Table 1.** Search parameters.

| Step 1 | |
| --- | --- |
| Maturity Model AND Chemical<br>Maturity Model AND Chemistry | ("Maturity Model" AND "Chemical") OR ("Maturity Model" AND "Chemistry") |
| **Step 2** | |
| ISO 17025 AND ISO 9001<br>ISO 17025 AND ISO 31000<br>ISO 17025 AND ISO 45001<br>ISO 17025 AND ISO 14001<br>ISO 17025 AND OHSAS 18001 | ("ISO 17025" AND "ISO 9001") OR ("ISO 17025" AND "ISO 31000") OR ("ISO 17025" AND "ISO 45001") OR ("ISO 17025" AND "ISO 14001") OR ("ISO 17025" AND "OHSAS 18001") |
| **Step 3** | |
| Maturity Model AND ISO 9001<br>Maturity Model AND ISO 17025<br>Maturity Model AND ISO 31000<br>Maturity Model AND ISO 45001<br>Maturity Model AND ISO 14001<br>Maturity Model AND OHSAS 18001 | ("Maturity Model" AND "ISO 9001") OR ("Maturity Model" AND "ISO 17025" OR ("Maturity Model" AND "ISO 31000") OR ("Maturity Model" AND "ISO 45001") OR ("Maturity Model" AND "ISO 14001") OR ("Maturity Model" AND "OHSAS 18001") |
| **Step 4** | |
| Maturity Model AND OHS<br>Maturity Model AND QMS<br>Maturity Model e Sustainable Development | ("Maturity Model" AND "OHS") OR ("Maturity Model" AND "QMS") OR ("Maturity Model" AND "Sustainable Development") |
| **Step 5** | |
| Maturity Model AND Integrated Management Systems | ("Maturity Model" AND "Integrated Management Systems") |

To define the target population, we chose laboratories that are in higher education institutions, that do chemical analyses, work with research and development, and provide external services to society (public laboratories) and laboratories that are private legal personalities, certified to the ABNT NBR ISO/IEC 17025 [40] standard, and focus on chemical analyses aimed at both environmental assessments and the oil, gas, and derivatives industry.

A total of 217 laboratories belonging to 147 public higher education institutions were surveyed, based on Ordinance No. 378, of 9 May 2016 [44], which establishes the list of units that make up the Federal Network of Professional, Scientific, and Technological Education [28]. In total, 545 private laboratories were also selected, using the register of the Brazilian Network of Testing Laboratories (RBLE) of the National Institute of Metrology (INMETRO). To select private laboratories, the following parameters were used: (1) Type of accreditation: "CRL (ABNT NBR ISO/IEC 17025—TEST LABORATORY)"; Test class: (2) "CHEMICAL TESTS"; (3) Areas of activity: "ENVIRONMENT" and "OIL AND DERIVATIVES, NATURAL GAS, ALCOHOL AND FUEL IN GENERAL".

In the third stage, a case study was carried out through a focus group to evaluate and validate the management practices formulated for a preliminary maturity model constructed from bibliographical research and questionnaire responses. The focus group included the participation of managers and technicians from a chemical analyses laboratory belonging to a public higher education institution. Participants were asked to assess (1) the levels at which management practices were found in the maturity model and (2) the level of importance versus implementation difficulty level.

To evaluate the model, two scales were used:

- Level of importance: 1-LOW IMPORTANCE, 2-IMPORTANT, 3-VERY IMPORTANT.
- Implementation difficulty level: 1-VERY EASY, 2-EASY, 3-MODERATE, 4-DIFFICULT, 5-VERY DIFFICULT.

The Focus Group's perceptions, suggestions, and changes were analyzed, and they helped adapt the final version of the model. The fourth and final stage of the work was the development of the maturity model. Based on the guide for developing maturity grids proposed by Maier [45], the foundations of the maturity model were developed for laboratory application.

Based on the articles researched in the literature review, the answers provided by managers through the questionnaire and the validation of the practices of the preliminary model by the focus group, it was possible to develop a maturity model for the gradual implementation of management systems, considering requirements related to technical skills for calibration and testing laboratories (ISO/IEC 17025 [40]), Quality Management (ISO 9001 [11]), Occupational Health and Safety Management (ISO 45001 [42]), Environmental Management (ISO 14001 [41]) and Risk Management (ISO 31000 standard [12]).

The model's target audience was managers, coordinators, and technical managers of testing laboratories (chemical analyses) located in public higher education institutions. As a framework, its objective was to inform laboratories of which steps can be taken to achieve excellence in management and sustainability.

## 3. Results and Discussion

### 3.1. Bibliographic Research

Bibliographic research clarified that higher education institutions (HEIs) are essential in disseminating information and in training professionals who will be attentive to sustainability issues. The role of HEIs is preponderant in sustainable development, as it promotes the development of actions to meet SDGs within the institution, raising awareness among professors, students, employees, and other interested parties and contributing to societal changes [46].

Since the establishment of the SDGs, many universities around the world have adhered to the topic, establishing policies and implementing actions that promote the sustainable development of campus activities [29,46]. To do this, they transform their missions, restructure their curricula, modify research programs, promote community engagement, and report their activities to stakeholders [47].

Linking the Institutional Development Plans (PDI) of Brazilian universities with the SDGs of the 2030 Agenda is a crucial step to be achieved. Serafini et al. [29] point out the following barriers to the implementation of SDGs in universities: the lack of documentation with standardized processes, the lack of training related to the SDGs for the academic community, difficulty in incorporating the SDGs into the institutional systems of HEIs; and cultural resistance to change.

Laboratories that have implemented management systems have their processes standardized and documented; this facilitates incorporation and alignment with the Sustainable Development Goals (SDGs) since it foresees the environmental aspects and impacts caused by their activities in their operations. Furthermore, they help implement the desired culture by minimizing resistance to imposed actions, as their technicians are constantly trained and are conscious of their role.

Teaching and research laboratories play a fundamental role in a country's economic and social development. Scientific and technological advancement and development significantly stem from research and experiments that have been tested, verified, and validated on laboratory benches. CONMETRO highlights the importance of chemical measurements to decision-making regarding product quality. Once the country is projecting itself as a protagonist on the world trade stage in the food, energy, and environment industries, it points to the need to immediately increase the reliability of the results of the chemical measurements done in Brazil.

The structural adaptation of laboratories inserted in HEIs to meet emerging service demands involves improving the technical skills of their members, which must include aspects related to the quality of operational and management processes, procedures for the safety and health of technicians, students, teachers, and other users; and environmental prevention practices.

Many companies and organizations have implemented management systems emphasizing quality, environmental, and occupational health and safety management to deal with contemporary pressures and complexities [15,48]. As a way to achieve the sustainability of their operations, many have opted for standards such as ISO 9001 (Quality), ISO 14001 (environment), and OHSAS 18001 (occupational health and safety) [5,49]. A survey carried out with certified Brazilian companies showed that those who achieved better sustainability performances were those who invested in improving the integration of their systems [15].

Nadae et al. (2020) [48] analyzed the impact of integrated management systems on sustainability in four Brazilian companies and concluded that investment in management systems improved the performance of their economic, social and environmental aspects (triple bottom line), despite sustainability not having been the primary motivation for implementing IMS.

Concerning implementing management systems in laboratories, it was possible to verify discussions on the quality of management and operations. The articles dealt with the benefits of adopting the ISO 9001 and ISO 17025 standards [49,50], critical analysis [51,52], and even describing implementation steps [52–55]. Only one article addressed the impact of normative standards [11,40,41,43,56] on chemical analyses activities linked to science and technology [57]. No articles were identified within the laboratories that addressed the use of occupational health and safety standards (ISO 45001) or risk management standards (ISO 31000) independently or integrated with the others.

The popularization of the use of management standards brings with it the need to assess maturity in several areas [15]. Integration can occur at different levels and the literature does not identify a single, standardized methodology and so each organizations implements the one that suits it best [58]. Maturity models can help organizations know where and how far they are from achieving best practices.

Domingues [13] proposed a maturity model to compare integrated management systems at different levels, evaluating the maturity level of the systems and directing companies to higher maturity levels. The research was designed using medium-sized companies located in a part of the Portuguese territory as a reference, limiting the sample to aspects related to geographic location, the IMS typology standard, and sectors of activity. The testing or calibration laboratories were not objects of the study.

Furthermore, the model used ISO standards as a conceptual basis before the current versions (2015) and before Annex SL and the High-Level Structure were published. The model also uses the OHSAS 18001 standard as the basis of the Occupational Health and Safety management system, not the ISO 45001:2018 standard. Finally, the model does not include aspects related to the operation of laboratories in its scope.

Regarding the use of maturity models applied to laboratories, Belezia [34] proposes a model that aims to evaluate competence, impartiality, and operational consistency for testing and calibration laboratories via self-assessment based on the requirements of Standard ABNT NBR ISO/IEC 17025:2017, employing decision support methods. The model uses the requirements in the ABNT NBR ISO/IEC 17025:2017 standard and the ABNT NBR ISO 9004:2010 standard to establish the criteria and assessment levels of the maturity model. Despite its high relevance as an assessment tool, the proposed model only addresses the assessment of the requirements of the ABNT NBR ISO/IEC 17025:2017 standard, and the evaluation of occupational safety and health or environmental aspects is not part of its scope. These are essential aspects of building sustainable institutions.

Gerônimo [59] carried out a systematic bibliographic review (RBS) in which he sought to identify maturity models based on fuzzy logic to assess the degree of maturity of laboratories that had implemented the ISO 17025 standard. The principal findings cited

by the author to develop maturity models were that the PDCA cycle is adopted for the implementation of management systems, maturity models are built using standards such as ISO standards with fuzzy logic, and maturity is generally evaluated on a five-point scale.

Gerônimo [35] proposed a descriptive maturity model to evaluate integrated management systems based on standards ABNT NBR ISO/IEC 17025:2017, ABNT NBR ISO 14001:2015, and ABNT NBR ISO 45001:2018 for testing and calibration laboratories. The model proposes using multi-criteria decision support methods and fuzzy logic for evaluation. The model, however, is only validated for the requirements of the ABNT NBR ISO/IEC 17025:2017 standard. The author emphasizes that the work was limited to developing a descriptive maturity model without taking improvement actions that could take the laboratory to a higher maturity level. In this way, the proposed model is limited to providing a situational diagnosis of the IMS's maturity level, and the proposal of effective integration actions that may improve the IMS's maturity level is not part of its scope.

In summary, the articles found did not demonstrate the existence of a maturity model that would assist chemical analyses laboratories in the gradual implementation of integrated management systems to meet and maintain sustainable levels in their operations. In the development of the theoretical framework, one can observe the role of laboratories in the training and qualification of future professionals and the need to build mechanisms that can handle the demands of quality, risks, safety, and the environment acceptably existing in their activities.

One can also observe that integrated management systems present themselves as a possible solution for this development since they can deal with the complexity of existing variables and because there is a relationship between increased integration and improved sustainability performance. Maturity models offer the possibility of measuring and following up with the gradual evolution of these systems.

### 3.2. Field Research

For data collection, six rounds of emails were sent: the first one between November and December 2022, and the remaining ones between January and March of 2023. Participation in the research was voluntary (by adhesion), ensuring the confidentiality of participants. A total of 701 emails were sent, and 42 managers responded to the questionnaire, representing 6.0% of the emails sent. The results described and the observations, analyses, and discussions presented were based only on the laboratories that participated in the survey. The research showed that the laboratories surveyed have lean structures, mainly operating with up to 20 employees (Figure 2).

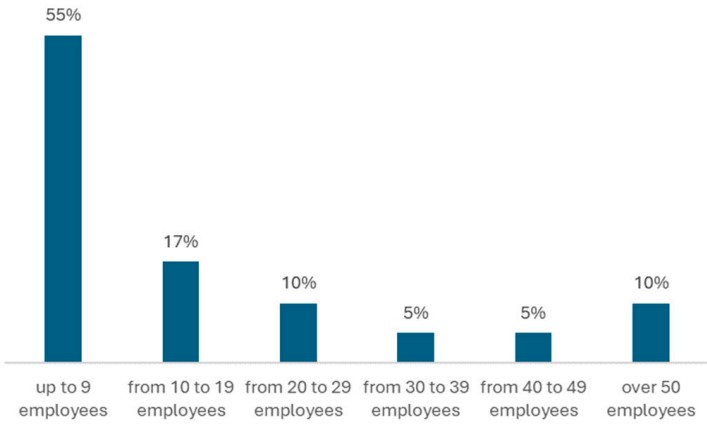

**Figure 2.** Number of laboratory employees.

The number of laboratory employees may be considered an impediment to adopting one or more normative systems. Barradas and Sampaio [52] observe that, among the benefits of adopting standard ISO 17025, there is an increase in the number of clients and the resulting increase in the laboratory's workload, which reflects customer's demands for

accredited calibrations. This could represent a need to expand the workforce, depending on an analysis of the return on the investment.

It can also be observed that almost half of the laboratory managers (47.6% of private laboratories and 42.9% of public laboratories) are not aware of the existing maturity models. The most cited model is the ABNT NBR ISO 9004 standard [20]—Quality management—Quality of an Organization—Guidance to achieve sustained success (Figure 3).

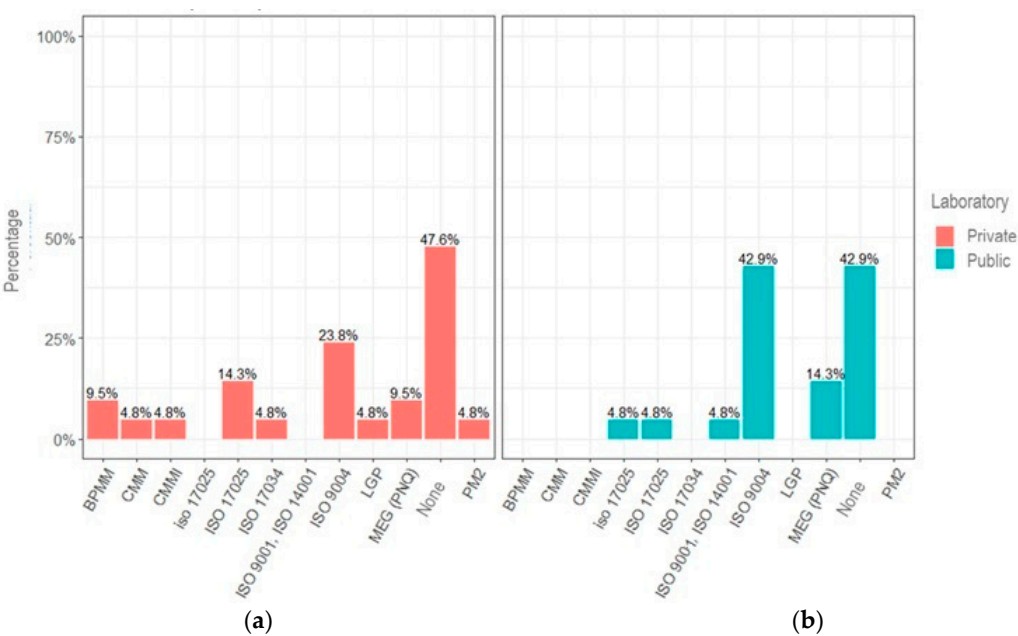

**Figure 3.** Maturity models that are known by laboratory managers. (**a**) Private laboratories; (**b**) Public laboratories [11,20,40,41,60].

The ISO 17025 standards [40] regarding laboratory accreditation and ISO 17034 [60] are worth mentioning, which define the requirements for producers of reference material when asked about knowledge of a maturity model applied to the laboratory setting. Both standards provide high standardization of operations and documents but do not establish gradual stages of evolution, allowing the laboratory to see the level of maturity at which it finds itself. The answer may demonstrate a lack of knowledge of the conceptual bases of a maturity model.

Regarding the relevance of a maturity model, 81% of private and public laboratory managers agree that a model would help improve internal and external processes (Figure 4). This vision is consistent with the context in which public managers find themselves, where the growing demand for the provision of quality public services has been the object of constant improvements in the internal processes of universities and continuous training of employees; this is an element that guides universities' Institutional Development Plans (PDI) [61].

Regarding market gain perception, a much greater variability of opinions was observed for public managers (24% partially disagree, and 33% see it as something indifferent to their reality). It is essential to highlight, however, that the understanding of market gains for laboratories within universities can translate into a higher success rate in the submission of projects tendered by private entities or even a demand for paid services via foundations, both arising from an acknowledgment of the technical competence of the laboratory.

When evaluating the adoption of integrated management systems, research revealed that public managers, for the most part, understand that they optimize laboratory processes (question 3.1.1); they can contribute to the hiring of new services (question 3.1.2) which do not negatively impact the time and way in which tasks are performed (question 3.1.3); and they understand that accreditation is relevant to improving the quality of operations (ques-

tion 3.1.7). The responses pointed to greater acceptance of the use of certified management systems by public managers when compared to private laboratory managers. Figure 5 presents the commented results.

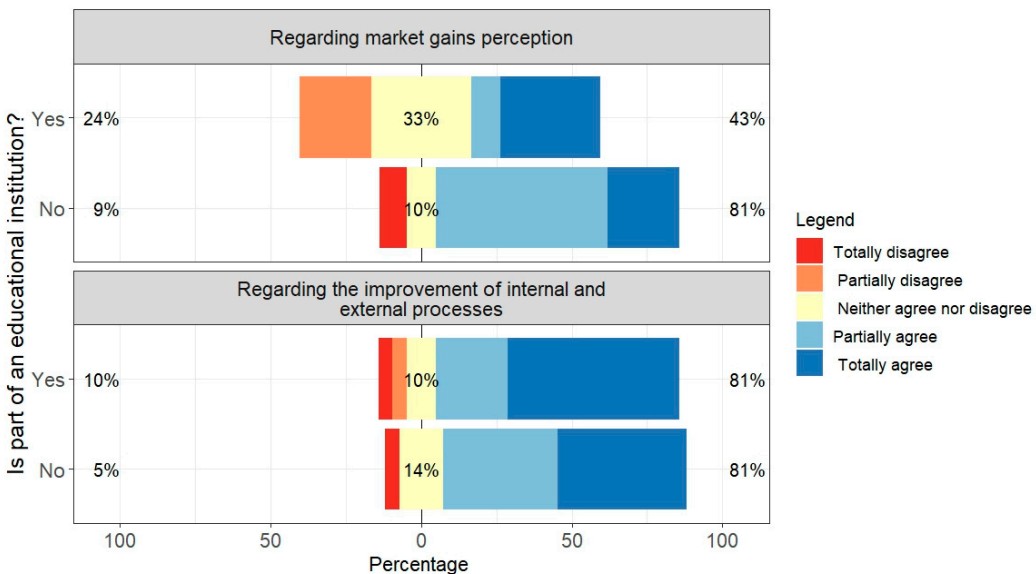

**Figure 4.** Perception of the relevance of adopting a maturity model.

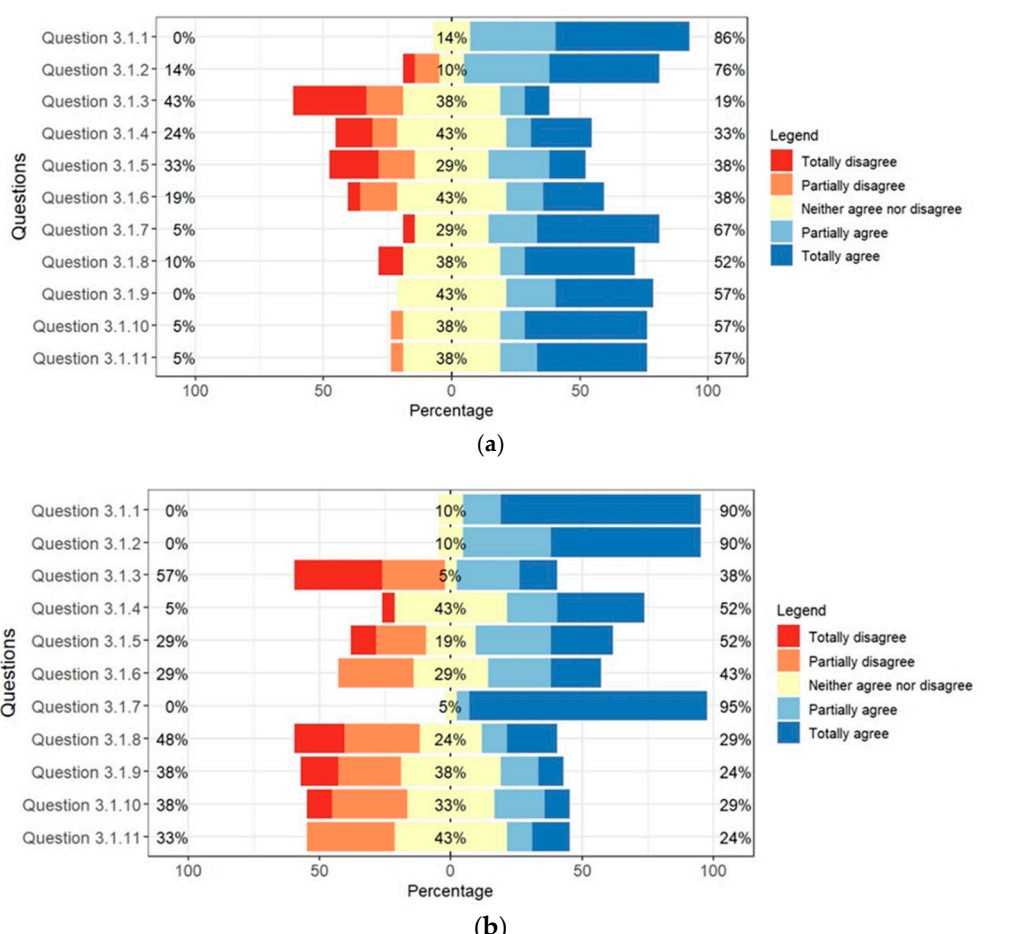

**Figure 5.** Classification of requirements according to relevance for the adoption of management systems (**a**) Public laboratories; (**b**) Private laboratories.

The ISO 17025 standard aims to promote confidence in the operation of laboratories and contains requirements that allow it to demonstrate that laboratories operate competently and can generate valid results [40]. According Sampaio and Barradas [52], the accreditation of a laboratory further increases the organization's performance through a better control of laboratory procedures, thus improving its potential due to increased customer satisfaction.

The research indicated that 67% of the laboratories surveyed, belong to public educational institutions and are not yet accredited based on the ISO 17025 standard (Figure 6). Choosing accreditation involves carefully assessing internal and external processes and, equally, an economic feasibility study that guarantees the sustainability of operations. The numbers presented, however, highlight the potential to be explored by public higher education institutions.

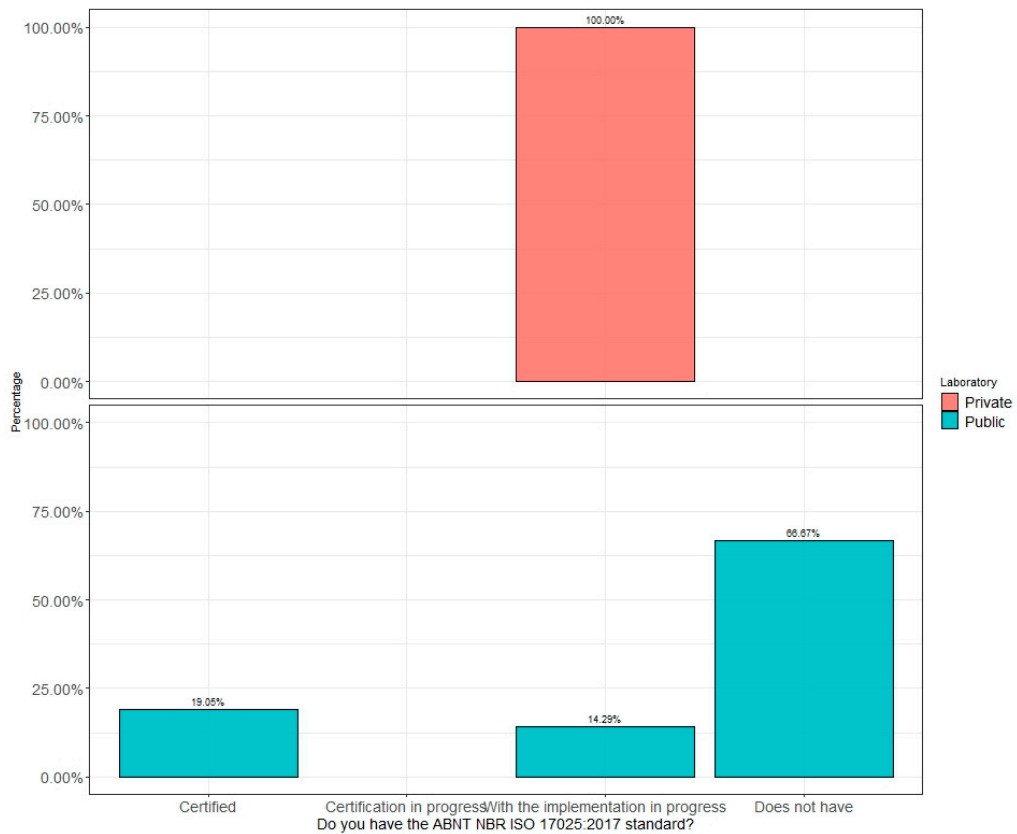

**Figure 6.** Adoption of the ISO/IEC 17025 standard [40] by laboratories.

Among the difficulties reported by public managers in implementing the standards, there is a lack of human resources (90.5%), followed by a lack of financial resources (71.4%), and thirdly, a lack of infrastructure (42.9%). A particular aspect of laboratories inserted in HEIs refers to the restrictions imposed by the regulations of public institutions. (38.1%) (Figure 7).

Unlike private companies, public institutions base their activities on legislation founded on ordinances, standards, and resolutions that public managers must abide by. The significant number of standards and regulations may impact execution time or make specific projects unfeasible in public institutions.

Thus, it is necessary to know the content of standards and regulations (legal requirements), know bureaucratic processes, and map management positions responsible for decision-making (stakeholders) to minimize the time impact of the adjustment necessary to the implementation process.

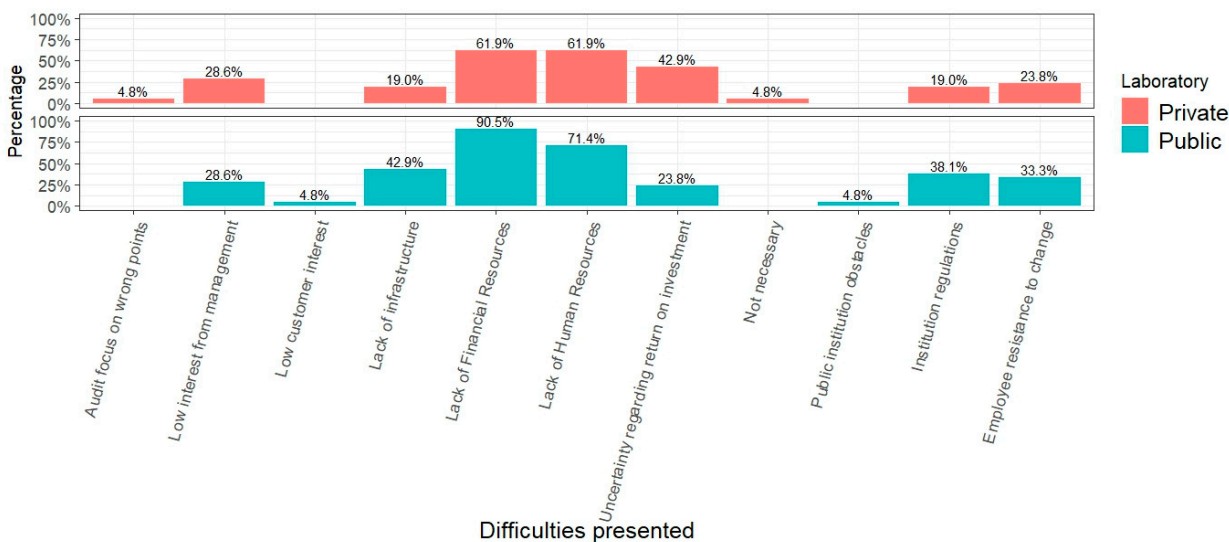

**Figure 7.** Reasons that make it difficult to implement one or more management systems.

Another verified aspect was the use of practices and documents necessary to manage the system well, per the requirements of ISO standards. Figure 8 presents the survey carried out for the ISO/IEC 17025 standard for public and private laboratories.

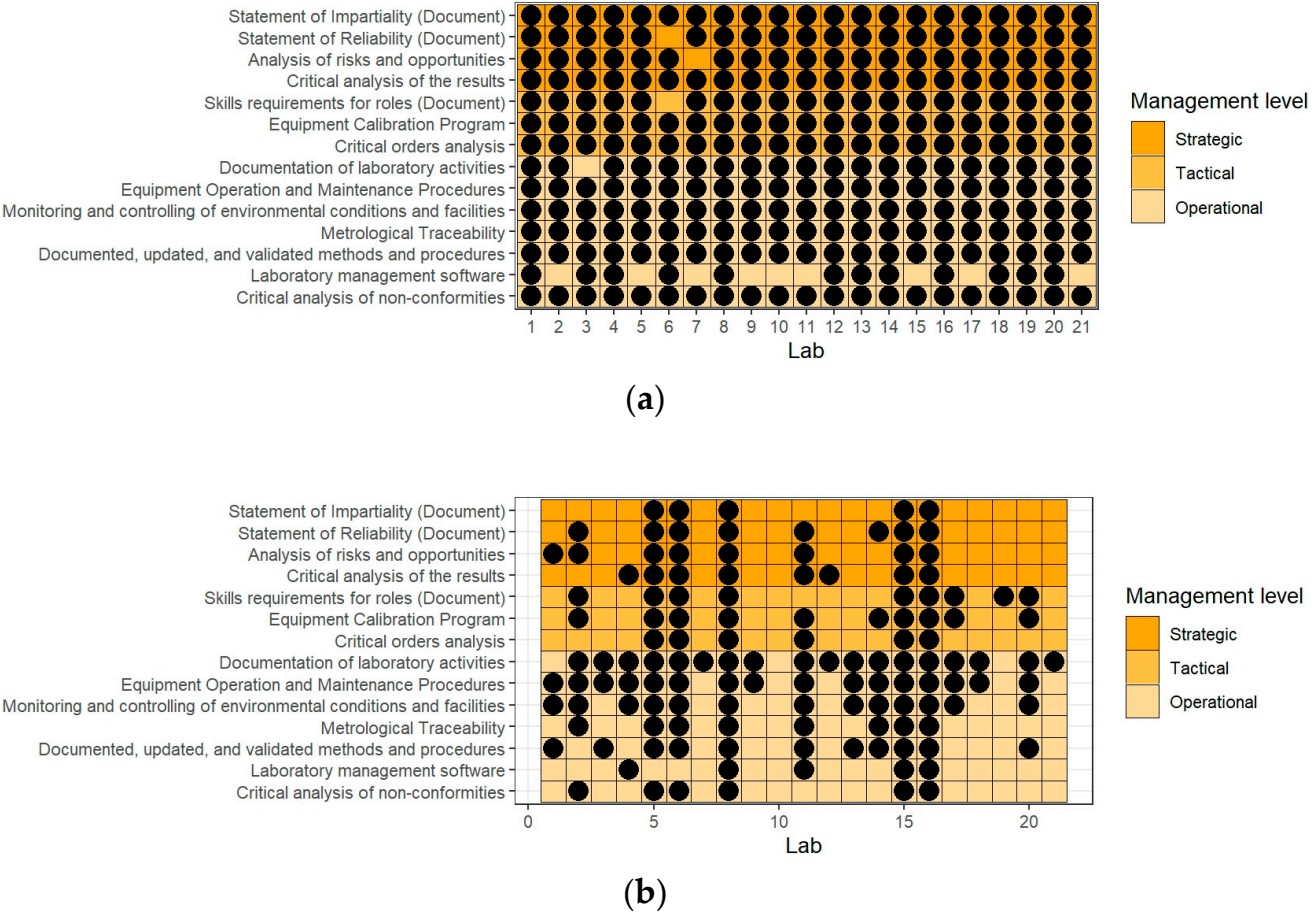

**Figure 8.** Practices, tools, and methods used in laboratory management classified from a strategic, tactical, and operational perspective. (**a**) Public laboratories; (**b**) Private laboratories.

Classified according to a hierarchical perspective of strategic, tactical, and operational management, it is clear that private laboratories have a set of practices and a much more homogeneous documentary structure than public laboratories. Specifically, concerning public laboratories, there is a predominance of more operational management practices and a smaller amount of tactical or strategic practices. Similar situations were observed in the analysis of the other ISO standards that were the subject of this study.

Notably, using these documents allows planning guided by clear objectives and goals, supported by methods and tools that will enable gradual monitoring of activities, with a view to continuous improvement of services. In their structure, they have logically interrelated elements, connecting management mechanisms and providing a feedback cycle that contributes to the sustainability of the laboratory.

### 3.3. Case Study (Focus Group)

The case study revealed the most challenging management practices to implement (in the focus group's view) and the need for more than one level in the maturity model (Level 0) to adapt a laboratory's activities before starting practices leading to accreditation.

Table 2 presents the validation of practices for the "Strategy" dimension. For each management practice, suggested levels of preparation and implementation of the preliminary maturity model were presented. The validated level represents the focus group's view of the positioning of practices within the model.

**Table 2.** Validation of practices in the "Strategy" dimension.

| Dimensions | Subdimension | Management Practices | | First Version Level | | Validated Practice | Validated Level | |
|---|---|---|---|---|---|---|---|---|
| | | | | MM Level | Preparation Level | | MM Level | Preparation Level |
| 1. Strategy | Initial diagnosis | PE 1.5 | The laboratory has identified stakeholders' quality, safety, and environmental requirements. | N5 | N2 | ok | ok | N1 |
| | | PE 1.8 | The laboratory recognizes the Hazards and Safety Risks of its operations. | N5 | N2 | ok | ok | N1 |

The focus group also revealed different perceptions between the management and technical teams regarding implementation difficulties and the importance of some practices. Figure 9 presents the analysis carried out for the "Strategy" dimension.

Among the implementation difficulties pointed out by the focus group in laboratories belonging to HEIs, the following are mentioned:

- Fellows' length of stay in projects: the short length of stay and high turnover affect service performance as there is a need for recurrent training of new members;
- Multipurpose laboratories (teaching, research, and services): in multipurpose laboratories, there is difficulty in implementing access controls necessary to comply with the ISO 17025 standard;
- Infrastructure adequacy: Some laboratories within universities need structural adjustments that make it challenging to establish material transport flows, another requirement of standard ISO 17025.

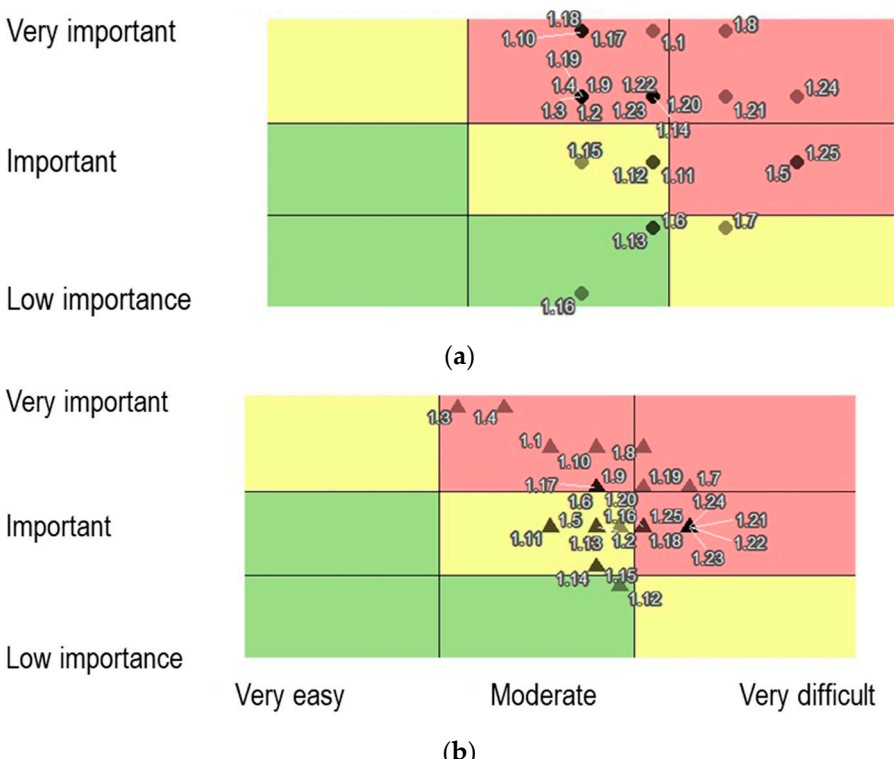

**Figure 9.** Importance vs Difficulty matrix of implementing the "Strategy" dimension. (**a**) Management team; (**b**) Technical team.

### 3.4. Maturity Model

As previously mentioned, public HEIs are kept and funded by the government and are subject to budgetary restrictions that impact the management and maintenance of the institution's infrastructure. Due to this, the construction of a maturity model had to consider some assumptions to adapt to this context.

Field research revealed that, on a scale of priorities, ISO 17025 and ISO 9001 are the most relevant standards for laboratories. ISO standard 17025 lays out the necessary parameters for laboratories to operate competently, and adopting requirements related to standard ISO 9001 makes process management improvement possible, which ensures laboratory quality results. For Domingues [13], when there is a quality management system already implemented, then the IMS implementation become less bureaucratic.

Element integration in management systems is a gradual process that occurs at several levels, and each level has different characteristics in terms of document, resource, and procedure integration. In the case of public HEIs, maturity models have to consider gradual implementation and integration of management systems, seeing that resource constraint is one of the factors mentioned that make implementation more difficult. As proposed by the focus group, model implementation using six maturity levels was also considered. Creating Level 0 resulted from adjustments to the levels of preparation of management practices arising from the experience of the laboratory's management and operations team. Many laboratories in public HEIs would initially require a level of adjustment to start the systems' implementation.

Field research revealed that the ISO 9004 standard is the best-known model for participants. Domingues [13] refers to the fact that this standard is a generic model, allowing for requirements to be set up according to the organization's specific needs. The fact that its application is based on a self-assessment makes some authors question whether or not it is feasible. However, being familiar with the terms of the normative requirements and the ISO standards structure facilitates understanding and model acceptance.

Domingues [13] adds that a risk management approach should accompany an IMS implementation. This approach takes the focus off of the quality management system (QMS), equalizing the importance of the other IMS norms. The proposed model was structured considering the requirements found on the ISO 31000 standard.

Another important aspect refers to systems' accreditation and certifications. Laboratory accreditation is desirable even in the public HEIs context. Barradas and Sampaio [39] clarify that the main reason for accreditation is related to market requirements, gauging service providing to external customers, and acknowledging quality services by countries that have signed cooperation agreements. Accreditation also contributes to the improvement of work practices, actions based on trustworthy results, and agility, safety, and the use of quality controls to maintain process efficacy and efficiency [47].

Belezia [34] lists the following advantages regarding accreditation: increasing the number of clients; increasing client satisfaction; improving facilities; improving training and personnel engagement; improving process management; improving activity cost. Despite the advantages reported, some difficulties for implementation are equipment management and difficulty in estimating uncertainties [52], as well as accreditation-related negative impacts such as increasing bureaucracy and requirements for time and budgets [34].

Barradas e Sampaio [52] warn that the internal high costs and lack of interest in providing external services are considered relevant reasons to give up on implementation. The latter option for integrating new management systems, however, reduces the maintenance cost of isolated systems in that it uses existing structures from its predecessors [9].

This way, amongst the assumptions to build a maturity model, a choice was made for the non-compulsory accreditation or certification of systems for level change because they would demand periodic maintenance costs. The decision for certification should consider a cost-benefit analysis of services offered by the laboratory.

Field research revealed that using management systems in university laboratories provides a greater perception of relevance for improving internal and external processes than possible market gains. This perspective is consistent with the context in which public managers find themselves, i.e., one in which the increasingly growing demand for the provision of quality public services, which has been the object of constant improvements in the internal processes of universities and continuous training of employees as it is a guiding element for the HEIs' Institutional Development Plans (PDI) [61].

It is essential to highlight, however, that the understanding of market gains for laboratories within public HEIs can translate into a higher success rate in the submission of projects tendered by private entities or even a demand for paid services via foundations, both arising from an acknowledgment of the technical competence of the laboratory.

Lastly, the proposed maturity model considers that the management system that results from the implementation should have fully integrated documentation. According to Jørgensen [4], for a company to be on a path towards sustainable management, it is necessary to focus on integrating different management standards. Poltronieri, Ganga e Gerolamo [15] have observed that the certified Brazilian companies that have obtained better performances on sustainability were the ones that invested in improving their systems integration.

Nadae et al. [48] analyzed the impact of integrated management systems on sustainability in four Brazilian companies and concluded that investment in management systems improved the performance of their economic, social and environmental aspects (triple bottom line), despite sustainability not having been the primary motivation for implementing IMS.

Integration can happen at different levels. According to Bernardo [58] they can be grouped into three: no integration, partial integration, full integration. No integration implies that the management systems (MS) implemented are managed separately; partial integration means that some elements of the MSs are common; and full integration indicates that all the elements of all the MSs are managed as an IMS. Maturity models can help organizations know where and how far they are from achieving best practices [15].

Table 3 summarizes the discussions regarding the assumptions to build a maturity model.

**Table 3.** Assumptions in building the maturity model.

| Assumptions | Justifications |
|---|---|
| Use the ISO standards structure to build the maturity model. | Field research revealed that the ISO 9004 standard is the best-known model for participants. Affinity with the terms of the normative requirements and the ISO standards structure facilitates understanding and model acceptance. "All requirements in this Standard are generic and intended to apply to all organizations, regardless of their type, size, and the product and service they provide" [11] (p. 3). |
| Start implementation using the ISO 17025 standard, followed by ISO 9001. | Field research revealed that they are the most relevant standards for laboratories on a scale of priorities. "The implementation of an IMS is less bureaucratic when companies already have at least one QMS implemented [. . .]" [13] (p. 34). |
| Deployments must occur in a sequenced manner (Step-by-Step). | Field research revealed that the laboratories' main complaints refer to resource constraints. |
| Implementing the requirements in the ISO 31000 standard must be carried out in conjunction with the ISO 9001 standard. | "[. . .] a risk management approach must accompany the implementation of an IMS, this being the integrating factor and the OHSMS being the pivot management subsystem, removing the focus from the QMS. Implicitly or explicitly, risk analysis is present in all subsystem references." [13] (p. 23). |
| There is no certification requirement to change levels. | The standards are not compulsory but operate by adhesion. Certification requires periodic maintenance of certificates. The decision for certification should consider a cost-benefit analysis. "[. . .] when systems are implemented in an integrated way and within a strategic and systemic vision, benefits are increased because processes are optimized [. . .])". [13] (p. 34). |
| Management systems with fully integrated documents. | Improving system integration contributes to Sustainability and points to building ways to bring system maturity to higher levels. [15]. Investment in management systems improved the performance of companies' economic, social, and environmental aspects (TBL) [48]. |
| Preferred focus on internal and external processes. | Field research revealed that using management systems in university laboratories has a more excellent perception of relevance for improving internal and external processes. "Internal or predominantly internal motivations are the "driving force" that leads companies to integrate their management subsystems" [13] (p. 24). |
| The model has 6 (six) maturity levels. | Creating Level 0 resulted from adjustments to the levels of preparation of management practices arising from the experience of the laboratory's management and operation team. |

The construction of the dimensions and sub-dimensions of the proposed model was carried out through a critical analysis of the process areas used in the maturity models and the integration models researched in the literature review.

The model was built with eight dimensions and 41 sub-dimensions, with an emphasis on systems integration to achieve sustainability of laboratory operations and based on the following success criteria: performance strategy clarity, leadership commitment; management excellence, operations reliability; continuous improvement systems integration to promote sustainability; and strengthening the culture in IMS (Figure 10).

To build the model's management practices, we observed the requirements of ABNT NBR ISO/IEC 17025:2017 standards; ABNT NBR ISO 9001:2015; ABNT NBR ISO 14001:2015; ABNT NBR ISO 45001:2017; ABNT NBR ISO 31000:2018, in addition to the GRI (Global Report Initiative) standards, which are all internationally acknowledged as a set of good practices in their respective fields of work. These practices were classified into the eight dimensions of the conceptual model based on a critical analysis of normative requirements (ISO), grouping them into related categories [14]. Subdimensions were systematized according to the aggregation of these practices [62].

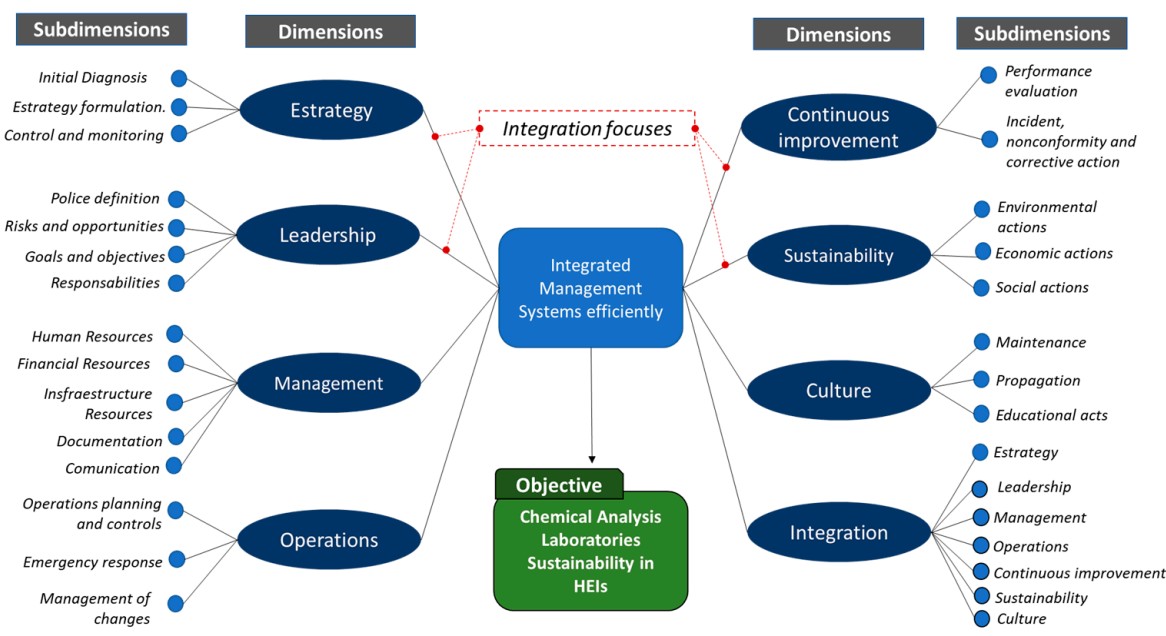

**Figure 10.** Concept diagram of the maturity model.

Table 4 presents management practices referring to the "strategy" dimension and the level at which this practice must be fully met. The level of preparation refers to the stage at which the laboratory starts the necessary adjustments for subsequent compliance with management practice.

**Table 4.** Management Practices of the maturity model of the "strategy" dimension.

| Dimensions | Subdimension | Management Practices | | | MM Level | Preparation Level |
|---|---|---|---|---|---|---|
| 1. Strategy | Initial diagnosis | PE | 1.1 | The laboratory recognizes the external and internal factors that may offer risks and opportunities for its work to continue. | N3 | N2 |
| | | PE | 1.2 | The laboratory has its macro processes mapped and linked to the value chain. | N3 | N2 |
| | | PE | 1.3 | The laboratory has identified all stakeholders. | N3 | N2 |
| | | PE | 1.4 | The laboratory has an established organizational identity with a vision, mission, and values. | N2 | N1 |
| | | PE | 1.5 | The laboratory has identified the quality, safety, and environmental requirements of stakeholders. | N5 | N1 |
| | | PE | 1.6 | The laboratory recognizes the critical success factors for the quality of its operations. | N2 | N1 |
| | | PE | 1.7 | The laboratory recognizes its operations' environmental aspects and impacts based on the life cycle of its services. | N4 | N3 |
| | | PE | 1.8 | The laboratory recognizes the hazards and safety risks of its operations. | N5 | N1 |
| | | PE | 1.9 | The laboratory's strategic planning considers the institution's PDI guidelines. | N3 | N2 |
| | Strategy Formulation | PE | 1.10 | The laboratory considers the requirements of impartiality and confidentiality when formulating its strategy. | N2 | N1 |
| | | PE | 1.11 | The laboratory considers stakeholder requirements in formulating the strategy. | N2 | N1 |

The model has 6 (six) maturity levels: STARTED, SUITABLE, DIFFERENCIATED, MANAGED, INTEGRATED, MAINTAINED. At each level, an objective must be achieved, and a set of management practices must be followed so that the laboratory can evolve progressively (Figure 11).

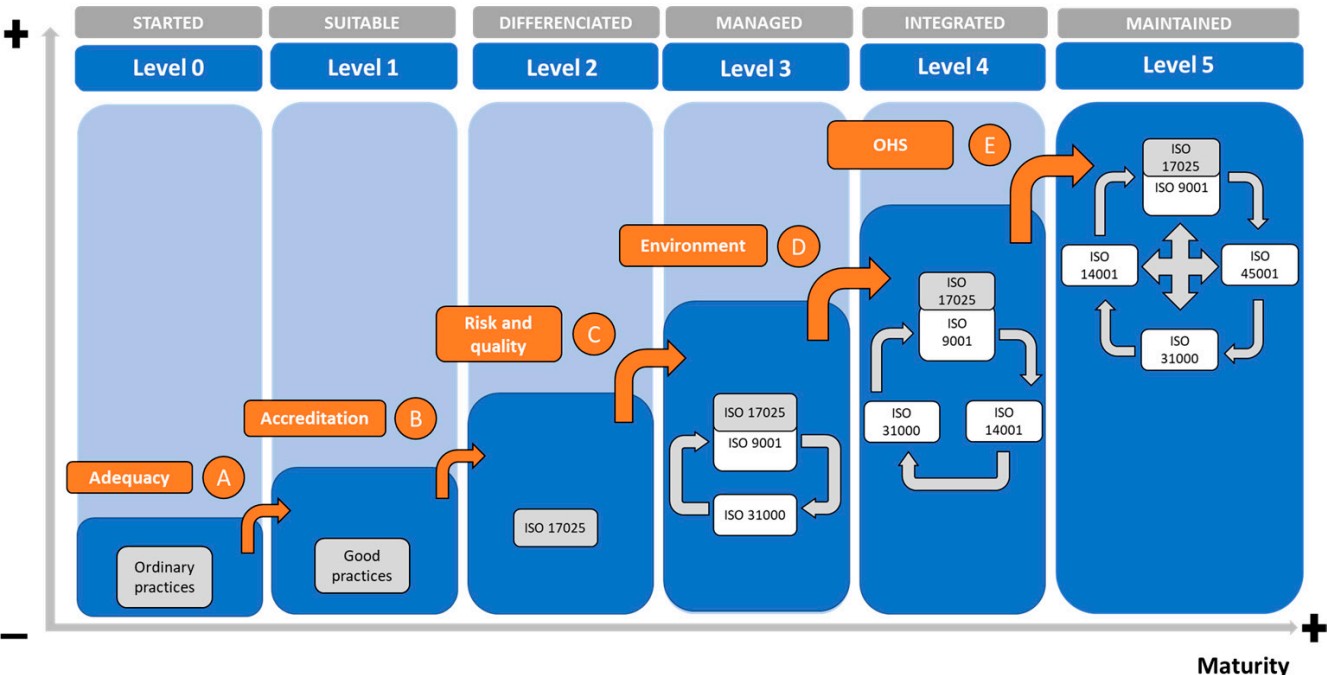

**Figure 11.** Evolution stages of the maturity model [11,12,40–42].

As an example, we cite the adequacy practices that the laboratory must adopt to move from Level 0 to Level 1: promote training to improve the skills necessary to carry out the activities of its technicians; adopt management, monitoring, and control techniques for the correct disposal of chemical products; adopt management, monitoring and control techniques to avoid contamination of discarded water; use of a chemical compatibility table for the correct storage of the chemical products used; extend your OHS operational controls to contractors and outsourced workers; and instruct people on how to proceed in emergency situations. Table 5 presents the characteristics expected to be observed in laboratories for the "Strategy" dimension when they reach the desired maturity level.

What is expected at Level 5 is to have a management system where the policies, manuals and procedures of the subsystems are integrated, audits are also carried out in an integrated manner, and the laboratory promote a culture of commitment and active participation from all those who use the laboratory facilities to maintain the implemented changes.

**Table 5.** "Strategy" Dimension of the maturity model.

| Dimensions | STARTED Level 0 | SUITABLE Level 1 | DIFFERENTIATED Level 2 | MANAGED Level 3 | INTEGRATED Level 4 | MAINTAINED Level 5 |
|---|---|---|---|---|---|---|
| 1. Strategy | At this level, the laboratory is at the beginning of its operations. It has no defined strategic actions, just routine operations. | At this level, the laboratory has not yet defined strategic actions, but it is starting to develop its mission, vision, and values. It also initiates the recognition and involvement of interested parties in the implementation and (subsequent) integration processes, the analysis of the critical success factors, and acknowledges OHS risks and hazards in its operations. | The laboratory's operational activities are based on the ABNT NBR ISO/IEC 17025 standard. It also has a mission, vision, values, critical success factors, requirements for reliability and impartiality, and stakeholders linked to its strategy. At this level, the laboratory starts (among other activities) to recognize external and internal factors that affect it, links the macro processes mapped to its value chain, and integrates systems beginning at the highest level. | At Level 3, the laboratory is certified (or operates) according to the ABNT NBR ISO 9001 standard, is accredited according to the ABNT NBR ISO/IEC 17025 standard, and operates according to the requirements of the ABNT NBR ISO 31000 standard. At this level, the laboratory considers aspects of quality and risk when formulating its strategy and has strategic indicators linked to microprocessors. At this level, the laboratory also begins to survey the environmental aspects and impacts of its operations based on the life cycle of its services. It begins to develop its environmental management system. | At Level 4, the laboratory is accredited (or operates) according to the ABNT NBR ISO/IEC 17025 standard. It is certified or operates according to standards ABNT NBR ISO 9001, ABNT NBR ISO 31000, and ABNT NBR ISO 14001. The laboratory recognizes its operations' environmental aspects and impacts and considers environmental aspects when formulating the strategy. At this level, the laboratory uses the recognition of hazards and risks to begin implementing the OHS management system. | At Level 5, the laboratory is accredited according to the ABNT NBR ISO/IEC 17025 standard. It is certified or operates according to standards ABNT NBR ISO 9001, ABNT NBR ISO 31000, ABNT NBR ISO 45001, and ABNT NBR ISO 14001. The laboratory recognizes the hazards and risks of its operations and considers OHS aspects when formulating its strategy. |

## 4. Conclusions

### 4.1. Conclusions

The maturity model (framework) aims to help chemical analyses laboratories located in public higher education institutions achieve higher levels of excellence in management and sustainability. To achieve this objective, concepts and structures from existing maturity models were linked to principles, guidelines, and requirements of standardized standards since they constitute a guide to best management practices. In addition, field research and a case study were carried out to understand whether it was relevant to the target audience.

As reported in the research, maturity models can help laboratories achieve higher levels of excellence by classifying the characteristics that exist at each maturity level and explaining the actions necessary to reach the highest levels.

Maturity models found in this work's development aimed to assess systems based on the decision-support methods, and only one model considered requirements related to safety and the environment. It is not part of the scope of these models to propose actions to the laboratory managers to increase or support maturity levels that had been reached; they were only offering a diagnosis of the situation.

The model proposed in this paper was structured to consider quality, safety, environmental standards, and aspects related to risk management and sustainability. From its onset, it has considered limitations and problems laboratories in Brazilian public HEIs have, being built in a step-by-step structure of management systems implementation and pointing at what practices should be done to meet superior levels.

Based on the propositions presented, it was observed that:

P1: The relevance of a management systems-based model was demonstrated in the theoretical framework and the field research carried out. More than 80% said they believed that a maturity model would help the organization of internal processes, with divergences only regarding market gains.

P2: The interest of managers can be observed in that 86% of public laboratory managers understand that using management systems optimizes laboratory processes and can contribute to hiring new services (Figure 5).

P3: Chemical analyses laboratories in universities and HEIs do not have structured management systems, relying solely on isolated tools and/or methodologies to coordinate and control their operations and manage routines. This proposition was confirmed when it was verified that the methods and tools were present in laboratory activities without structuring elements that would configure a management system (Policy, Objectives, Goals, Manual, and other documents) (Figure 8).

P4: It was possible to structure a maturity model suited to the context of chemical analyses laboratories to achieve sustainable activities. This proposition was confirmed by structuring the maturity model presented in Figures 10 and 11.

The topic is deemed relevant for laboratory managers. It has an immense field to be explored, as many research participants say they do not have fully implemented management systems, observing only isolated tools and methods. Structural and managerial suitability of laboratories in public HEIs based on standards would allow for the use of the same document and organizational standard and of using a language similar to the one used in businesses and private laboratories, contributing to demands that emerge from services in tender processes and occasional decision-making in partnership agreements.

When analyzing the public laboratories researched based on the maturity model developed, it can be observed that the majority of laboratories would be found at Levels 0 or 1 (67% are not accredited in the ISO 17025 standard and do not have a complete documentary structure, which is necessary for sound systems management). The difficulties reported by public managers in implementing the standards to a certain extent point to the reasons for the low level of maturity: lack of human resources (90.5%), financial resources (71.4%), and infrastructure (42.9%).

Comparatively, private laboratories surveyed could be classified in Levels 2 or 3, depending on their compliance with the quality management and risk management criteria present in the IMS (only 14.3% of private laboratories stated that they had ISO 9001 certification or were in the implementation process, and 4.7% for risk management).

It is therefore noted that the maturity model could help laboratories achieve higher levels of sustainability, as it indicates the necessary actions to be taken to gradually evolve their operations and management practices. Furthermore, the model contributes to the implementation of actions for the sustainable development of public HEIs, aligning the institution's strategic mission with the Sustainable Development Goals (SDGs) from the 2030 agenda.

### 4.2. Countermeasure

Chemical analysis laboratories in HEIs have significant environmental and operational risks that should not be neglected. Manipulating toxic, irritant, asphyxiating, explosives or flammable chemicals and engaging in experiments exposes technicians, professors, and students to the most diverse risk situations.

In this way, laboratories need safety and emergency programs from the start of their activities to maintain the integrity of their facilities and users and ensure operational sustainability. In this reality, establishing standards, procedures, and safety and emergency management mechanisms at the first levels of implementation of the proposed maturity model is recommended.

### 4.3. Limitations

Laboratories serve different purposes, and the building of their environment will depend on the problem they are destined to treat. The model was developed based on answers from chemical analysis laboratory managers in the "environment" and "oil and derivatives, natural gas, ethanol and fuels in general".

Six rounds of sending emails were carried out. From the 2nd round of sending onwards, telephone calls were also made. Some emails were lost in spam boxes, and in these situations where the individuals were identified through telephone contact, people were not interested in participating in the research. Therefore, the number of responses to the field research did not allow statistical inference of the data.

### 4.4. Suggestions for Future Researchs

By conducting a process of gradual implementation of integrated management systems, the model also collaborates with the implementation of sustainability policies and actions within higher education institutions (HEIs), as laboratories with implemented management systems have their processes standardized and documented, which facilitates incorporation and alignment with the Sustainable Development Goals (SDGs), present in the Institutional Development Plans (PDI) of universities. Furthermore, they help to implement the desired culture by minimizing resistance to imposed actions as their technicians are constantly trained and made aware of quality, safety, and environmental aspects.

Based on what was developed, the following are recommended for future research:

- Develop a measurement scale for the maturity model based on methods such as multi-criteria decision;
- Validate the proposed maturity model based on the evaluation of educational institutions' laboratories (Benchmarking);
- Test and expand the model for laboratories with different characteristics from the research scope;
- Assessing the resilience of systems implemented for the maintenance of the laboratory sustainability;
- Evaluate the contribution of the proposed maturity model to the success of project submission (scientific production);
- Evaluate the costs of implementing one or more management systems based on the return on investment (ROI).

It is expected that the proposed model will contribute to the improvement of laboratory management, sustainable development of their activities, and quality of services provided to the community by public higher education institutions in the country.

**Author Contributions:** All the authors contributed to writing and editing the published version of the manuscript. All authors have read and agreed to the published version of the manuscript.

**Funding:** This research received no external funding.

**Institutional Review Board Statement:** Not applicable.

**Informed Consent Statement:** Not applicable.

**Data Availability Statement:** The data presented in this study are available on request from the corresponding author.

**Acknowledgments:** The authors would like to thank the Rio Grande do Norte Federal University (UFRN), Brazil, Department of Petroleum Engineering, Postgraduate Program in Science and Petroleum Engineering (PPGCEP).

**Conflicts of Interest:** The authors declare no conflicts of interest.

## Appendix A. Questionnaire Sent to Laboratory Managers

Section 1—Institutional Data

1.1. Respondent position/role: _______________________________________________

1.2. Year the laboratory was founded: _______________

1.3. Number of employees:

1.4. State: ___________

1.5. Educational institution: ☐ No ☐ Yes Name: __________________

Section 2—Maturity Models Perception

| A maturity model can be understood as a tool that helps companies understand the quality of their processes and establishes gradual transition stages until a level considered to be excellent is reached. |
| --- |

2.1. What Maturity Models do you know or have heard of?

☐ KMM          ☐ PM2          ☐ MEG (PNQ)
☐ None          ☐ MMGP         ☐ CMM
☐ Others:       ☐ PMM          ☐ CMMI
_________         ☐ BPMM         ☐ QMMG
                ☐ OPM-3        ☐ ISO 9004

2.2. Do you know of any maturity programs or models applied to the laboratory setting?

☐ Yes, which one? _______________
☐ No
☐ can't say

2.3. On a scale of 1 to 5, where 1 (STRONGLY DISAGREE) and (STRONGLY AGREE), classify the following requirements according to their relevance for management, operations, and maintenance of laboratory activities.

|       |                                                                                                                                             | 1 | 2 | 3 | 4 | 5 |
| ----- | ------------------------------------------------------------------------------------------------------------------------------------------- | - | - | - | - | - |
| 2.3.1 | A Maturity Model applicable to the laboratory setting helps guide the gradual adoption of management practices to improve internal and external processes. | δ | δ | δ | δ | δ |
| 2.3.2 | A Maturity Model applicable to the laboratory setting helps define differentiation strategies between competing laboratories, enabling market gains. | δ | δ | δ | δ | δ |

2.4. On a scale of 1 to 5, where 1 (OF LOW IMPORTANCE) and (VERY IMPORTANT), classify the following requirements according to their relevance for management, operations, and maintenance of laboratory activities.

| | | 1 | 2 | 3 | 4 | 5 |
|---|---|---|---|---|---|---|
| 2.4.1 | Identify all stakeholders in the laboratory's activities and define their requirements for quality of services. | δ | δ | δ | δ | δ |
| 2.4.2 | The laboratory recognizes the external and internal factors that may offer risks and opportunities for its work to continue. | δ | δ | δ | δ | δ |
| 2.4.3 | Acknowledging the Value Chain and its respective processes linked to strategic indicators. | δ | δ | δ | δ | δ |
| 2.4.4 | Definition of strategic information, processes, and stakeholders, selecting the most important ones for decision-making. | δ | δ | δ | δ | δ |
| 2.4.5 | Establishment of relationship channels with stakeholders to handle requests, complaints, and suggestions. | δ | δ | δ | δ | δ |
| 2.4.6 | Market analysis and segmentation. Definition of target customers and assessment of satisfaction, loyalty, and dissatisfaction. | δ | δ | δ | δ | δ |
| 2.4.7 | Identification, selection, qualification, and performance evaluation of suppliers. Performance communication. | δ | δ | δ | δ | δ |
| 2.4.8 | Selection and qualification of workers. Performance evaluation. | δ | δ | δ | δ | δ |
| 2.4.9 | Treatment of health and safety hazards and risks. Promotion of improved quality of life, well-being, and satisfaction. | δ | δ | δ | δ | δ |
| 2.4.10 | Identification of existing strengths and gaps in management. Definition of current and future competencies. | δ | δ | δ | δ | δ |
| 2.4.11 | Identification, development, retention, and protection of knowledge. | δ | δ | δ | δ | δ |
| 2.4.12 | Induction, development, and implementation of innovation. | δ | δ | δ | δ | δ |
| 2.4.13 | Identification of capacity for change, including assessment of need and capacity for implementation. | δ | δ | δ | δ | δ |
| 2.4.14 | Assessment of flexibility for changes, including review of strategies, goals, processes, and products at an appropriate time. | δ | δ | δ | δ | δ |
| 2.4.15 | Definition of values, principles, guidelines, and standards of conduct. Ethical relationship with stakeholders. | δ | δ | δ | δ | δ |
| 2.4.16 | Risk management, compliance with legal requirements, and transparency with stakeholders. | δ | δ | δ | δ | δ |
| 2.4.17 | Mapping of organizational culture to implement strategies and practice values. | δ | δ | δ | δ | δ |
| 2.4.18 | Performance analysis of indicators and monitoring of action plans and their resources | δ | δ | δ | δ | δ |
| 2.4.19 | Definition of leadership competencies and leader development. | δ | δ | δ | δ | δ |
| 2.4.20 | Defining and monitoring economic-financial indicators, cost management, budget, and fiscal control. | δ | δ | δ | δ | δ |
| 2.4.21 | Prevention, treatment, and monitoring of environmental impacts. Quick response to emergencies. | δ | δ | δ | δ | δ |
| 2.4.22 | Prevention, mitigation, and monitoring of social impacts. | δ | δ | δ | δ | δ |
| 2.4.23 | Implementation of information systems with the establishment of security requirements. | δ | δ | δ | δ | δ |
| 2.4.24 | Mapping, analysis, and improvement of laboratory processes. | δ | δ | δ | δ | δ |
| 2.4.25 | Identification of new product development opportunities | δ | δ | δ | δ | δ |

Section 3—Perception of Management Systems

Management Systems can be understood as interrelated and interdependent management practices and methods with pre-defined objectives and goals that help companies continuously improve in managing specific areas, such as Quality, Environment, and others.

3.1. On a scale of 1 to 5, where 1 (STRONGLY DISAGREE) and 5 (STRONGLY AGREE), classify the following requirements according to their relevance for management, operations, and maintenance of laboratory activities.

|  |  |  | 1 | 2 | 3 | 4 | 5 |
|---|---|---|---|---|---|---|---|
| 3.1.1 | | Adopting a certifiable Management System optimizes the laboratory's internal processes. | δ | δ | δ | δ | δ |
| 3.1.2 | | Adopting a certifiable Management System enhances the contracting of new services by the laboratory. | δ | δ | δ | δ | δ |
| 3.1.3 | | Adopting a certifiable Management System leads to rigid laboratory internal processes. | δ | δ | δ | δ | δ |
| 3.1.4 | | Adopting a certifiable Management System increases laboratory operating costs. | δ | δ | δ | δ | δ |
| 3.1.5 | | Adopting a certifiable Management System increases the complexity of laboratory management. | δ | δ | δ | δ | δ |
| 3.1.6 | | Adopting a certifiable Management System requires hiring more professionals to deal with the documents generated by the system. | δ | δ | δ | δ | δ |
| 3.1.7 | | Adopting the ABNT NBR ISO 17025:2017 standard to improve the quality of operations, compared to what we currently have, is relevant to the laboratory. | δ | δ | δ | δ | δ |
| 3.1.8 | | Adopting the ABNT NBR ISO 9001:2015 standard for implementing a quality management system for internal processes, compared to what we currently have, is relevant to the laboratory. | δ | δ | δ | δ | δ |
| 3.1.9 | | Adopting the ABNT NBR ISO 14001:2015 standard for implementing an environmental management system for internal processes, compared to what we currently have, is relevant to the laboratory. | δ | δ | δ | δ | δ |
| 3.1.10 | | The adoption of the ABNT NBR ISO 45001:2017 standard for implementing an occupational health and safety management system for internal processes, compared to what we currently have, is relevant to the laboratory. | δ | δ | δ | δ | δ |
| 3.1.11 | | Compared to what we currently have, adopting the ABNT NBR ISO 31000:2018 standard for implementing a risk management system for internal processes is relevant to the laboratory. | δ | δ | δ | δ | δ |

3.2. Does the laboratory adopt Laboratory Management practices, tools, and methods? Select the option corresponding to the practices adopted (you may select multiple answers).

☐ Statement of Impartiality (Document)
☐ Statement of Reliability (Document)
☐ Documentation of laboratory activities
☐ Skills requirements for roles (Document)
☐ Equipment Calibration Program
☐ Equipment Operation and Maintenance Procedures
☐ Monitoring and control of environmental conditions and facilities
☐ Metrological Traceability
☐ Documented, updated, and validated methods and procedures
☐ Laboratory management software
☐ Analysis of risks and opportunities
☐ Critical orders analysis
☐ Critical analysis of the results
☐ Critical analysis of non-conformities
☐ Others: ____________

3.3. Does the laboratory adopt strategic, tactical, and operational management methods? Select the option corresponding to the practices adopted (you may select multiple answers).

| | Quality | Environment | Occupational Health and Safety | Risks and Opportunities | Does Not Adopt |
|---|---|---|---|---|---|
| Management manual | δ | δ | δ | δ | δ |
| Policy | δ | δ | δ | δ | δ |
| Goals and objectives | δ | δ | δ | δ | δ |
| Written instructions and procedures | δ | δ | δ | δ | δ |
| Performance indicators | δ | δ | δ | δ | δ |
| Scheduled inspections | δ | δ | δ | δ | δ |
| Audits | δ | δ | δ | δ | δ |

3.4. Select the option that corresponds to the reality of the laboratory (you may select more than one answer). The laboratory has the standard…

| | Certified | Certification in Progress | With the Implementation in Progress | Does Not Have |
|---|---|---|---|---|
| ABNT NBR ISO 17025:2017 | δ | δ | δ | δ |
| ABNT NBR ISO 9001:2015 | δ | δ | δ | δ |
| ABNT NBR ISO 14001:2015 | δ | δ | δ | δ |
| ABNT NBR ISO 45001:2018 | δ | δ | δ | δ |
| ABNT NBR ISO 31000:2018 | δ | δ | δ | δ |
| Others: _______________ | δ | δ | δ | δ |

3.5. In your opinion, what reasons make it difficult to implement one or more management systems in the laboratory? (you may select more than one answer)

☐ Lack of Human Resources
☐ Lack of Financial Resources
☐ Lack of infrastructure
☐ Institution regulations
☐ Uncertainty regarding return on investment
☐ Low interest from management
☐ Employee resistance to change
☐ Others: ____________

Section 4—Adoption of Management Practices

4.1. Does the laboratory adopt Laboratory Management practices, tools, and methods? Select the option corresponding to the practices adopted (you may select multiple answers).

☐ Determining customer requirements
☐ Measuring customer satisfaction
☐ Others: ____________

☐ 5S Program
☐ FMEA
☐ Process Mapping
☐ Process Quality Control

4.2. Does the laboratory adopt Laboratory Management practices, tools, and methods? Select the option corresponding to the practices adopted (you may select multiple answers).

☐ 3R/5R Program

☐ Assessment of Environmental Aspects and Impacts

☐ Others: ____________

☐ Effluent treatment

☐ Conscious consumption (water/energy)

☐ Waste sorting

☐ Proper waste disposal

4.3. Does the laboratory adopt Occupational Health and Safety Management practices, tools, and methods? Select the option that corresponds to the practices adopted (you may select more than one answer).

☐ Good Laboratory Practices—GLP

☐ Chemical Compatibility Chart

☐ Labeling System

☐ PPE training

☐ Risk Management Program—RMP

☐ Environmental Risk Prevention Program—PPRA

☐ Occupational Health Medical Control Program—PCMSO

☐ Fire prevention and fire fighting measures

☐ Hazard and Risk Assessment

☐ Others: ____________

4.4. Does the laboratory adopt Laboratory Management practices, tools, and methods? Select the option that corresponds to the practices adopted (you may select more than one answer).

☐ Brainstorming

☐ Checklists

☐ Preliminary Risk Analysis—PRA

☐ Hazard Analysis and Critical Control Points—HACCP

☐ Failure Modes and Effects Analysis—FMEA

☐ Reliability Centered Maintenance—RCM

☐ Cause and Effect Analysis

☐ Probability and Consequences Matrix

☐ Others: ____________

4.5. The space below is intended for additional comments, criticisms, and suggestions regarding the questionnaire and/or about any matters pertinent to management systems and Maturity Models. We thank you in advance

Thank you for taking the time to answer this questionnaire. Thank you. (500 words)

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
