# Peer review of "Maturity Model for Sustainability Assessment of Chemical Analyses Laboratories in Public Higher Education Institutions"

_sustainability, doi:10.3390/su16052137_

Round 1

Reviewer 1 Report

Comments and Suggestions for Authors

The study 'Maturity Model for Sustainability Assessment of Chemical Analyses Laboratories in Public Higher Education Institutions' addresses the interesting topic of sustainability support in an adopted research area, worthy of scientific recognition. The authors focused on the framing of a maturity model based on standardised standards to guide adjustments related to the quality, risk, safety and environment of chemical analysis laboratories in public higher education institutions.  In this regard, the authors conducted their research in four sections: a theoretical study on maturity models, sustainability and integrated management systems; a research phase with laboratories; a case study in a university chemical analysis laboratory; and a section on constructing and validating the maturity model. 

The abstract of the article presents the background of the research and its subject matter. The stages of the research and an overview of the concept are presented. The abstract should indicate the research method and highlight more strongly the main results of the findings - in this respect the abstract should be refined.

The introductory section develops the research background signalled in the executive summary. It points to the directions of the management models being developed towards sustainability. I believe that this theme should be expanded to include management (maturity, business) models specific to the chemical industry. The above will more strongly justify maturity modelling at the level of university chemical laboratories. In this respect, it is worth supporting with a broader literature review (the reference of the article should be expanded). It is worth studying https://doi.org/10.3390/su15118889, https://doi.org/10.3390/su141811695, among others.

The introductory section presents the research assumptions, which should be more strongly related to the noted research gap.

The materials and methods section presents the research concept and research reference. Visualisation and tabulation facilitate the perception of the content.

The results and discussion section presents the findings according to the stages of research adopted. This is a well-done section of the study in terms of 'results'. However, a broader discussion of the literature is missing. I believe that the results section should be separated from the discussion section. The implications should be strengthened.

The summary section presents the main findings with reference to the research assumptions made. The practical significance of the findings should be more strongly emphasised. The novelty of the study in relation to the identified research gap should be pointed out.

The literature is quite scarce, I believe it should be expanded

In conclusion:

Supplement the summary with the research method and develop the main findings,

The research gap in response to which the article was written should be more strongly highlighted,

The introduction in terms of governance models specific to the chemical industry should be completed,

The results and discussion section should be separated and the discussion section should be refined,

The references should be strengthened.

Reviewer 2 Report

Comments and Suggestions for Authors

This article establishes a sustainable maturity model for chemical analysis experiments in higher education, which is well-structured and well-documented. Overall, this article might have been better if it had been modified in the following areas.

1. suggest that the introduction highlights the problem more, a good article needs to give the reader a clear idea of why the problem is being studied, the problem in this article is not exported directly enough. The structure of the introduction needs to be adjusted to emphasize the sense of the problem. For example, could paragraphs 1-3 be combined and narrowed? These paragraphs mainly introduce the larger context. They do not need to be too long; too long will lead to a less direct problem sense.

2. why a public school of higher education? What is special about public higher education? Is the model of this paper also applicable to private higher education institutions?

3. In the introduction, there are only 4 references, which is not enough to discuss the research frontiers of the scientific issues of this paper. The necessity of this paper is explained from the reality, but it is not enough to explain the necessity of this paper from the point of view of the inadequacy of the theoretical research, so it is recommended to strengthen it.

4. In the introduction, it is suggested to add a table to compare other scholars' sustainability evaluation models of laboratories, to check the advantages or to explain the shortcomings, and to further explain the necessity of this paper through comparison.

5. Figure 9 is pulled up vertically, which is not aesthetically pleasing, and it is suggested to modify it.

6. In the conclusion part, it is suggested to express in subsections, conclusion, countermeasure suggestions and limitations. A good research should have a guiding significance to the reality, and it is recommended to propose countermeasures for the sustainable management of laboratories based on the conclusions of the study, such as strengthening the investment in emergency management management (refer to the reference citation: https://doi.org/10.1016/j.jlp.2023.105230 ), because chemical laboratories have a certain degree of danger, and the investment in safety and emergency management is also very important to the sustainable development of the laboratory. The paper does not point out that this paper is not a good example of sustainable management.

7. This paper does not point out the limitations of this paper and future research direction, it is recommended to pay attention to the resilience of the relevant research, I think resilience and laboratory sustainability combined can be your next research direction.

8. Literature citation format, it is recommended to check the full text. There are low-level errors, such as literature [4] only year, volume, no month, issue, literature number 133473.

Round 2

Reviewer 1 Report

Comments and Suggestions for Authors

The authors have improved the article. Minor additions are still required - text needs to be completed between the table and figure (lines 626-624) and below the table - line 630.

References are still worth strengthening and editorial revision (e.g. line 126, 127).

Reviewer 2 Report

Comments and Suggestions for Authors

no

Author Response

We take this opportunity to thank you for your suggestions that contribute to improve the understanding of the article.

Thank you very much.

Marco Souza